# A Transportation Network Paradox: Consideration of Travel Time and Health Damage due to Pollution

**Zhaolin Cheng [1,2], Laijun Zhao [3,\*] and Huiyong Li [1]** 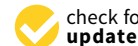

[1]   Sino-US Global Logistics Institute, Shanghai Jiao Tong University, 1954 Huashan Rd.,
     Shanghai 200030, China; 02273@zjhu.edu.cn (Z.C.); hyl_shu@163.com (H.L.)
[2]   Business School, Huzhou University, 759 Erhuandong Rd., Huzhou 313000, China
[3]   Business School, University of Shanghai for Science and Technology, 516 Jungong Rd.,
     Shanghai 200093, China
[\*]   Correspondence: ljzhao@usst.edu.cn

**Abstract:** In cities with serious air pollution, travel time and health damage significantly affect route choice by travelers (e.g., motorcycle and scooter drivers). Consequently, the classical Braess paradox is no longer realistic because it only considers the traveler's value of time (VOT). In the current study, we describe a new transportation network paradox that considers both the VOT and the traveler's perception of pollution damage. To examine the conditions that create the new paradox, we developed a novel method to compute a total comprehensive cost that combines the VOT with health damage. We analyzed the conditions for the new paradox and the system's performance using a user equilibrium model and system optimization. Furthermore, an improved model is used to analyze how different transport modes influence the Braess paradox. We found that whether the new paradox occurs and the potential improvement of the system's performance depend on whether the total travel demand falls within critical ranges. The bounds of these ranges depend on the values of the parameters in the function that describes the health damage and the link travel time function. In addition, high health damage significantly affects route choices and traffic flow distribution. This paper presents a new perspective for decision-making by transportation planners and for route choices in cities with serious air pollution.

**Keywords:** paradox; health damage; the value of time; user equilibrium; system optimization

---

## 1. Introduction

Air pollution has become an important topic in global environmental issues with economic development. A large number of greenhouse gases and pollutants are produced in the production, logistics, and other aspects of national economies. For example, large amounts of coal, coke, gasoline, diesel, and natural gas are consumed in manufacturing for the ferrous metal industry and nonferrous metal industries; $CO_2$ emissions are continuing to grow [1]. In seaborne trade, as emissions produced by oceangoing vessels and container handling equipment would lead to drastic climate changes, ship operators have to constantly adjust the speed and route to reduce air pollution [2,3]. The aviation industry is in a similar condition. According to the Air Transportation Action Group (ATAG), the global aviation industry produced about 705 million tons of $CO_2$ in 2013, which is about 2% of the total $CO_2$ emissions and 13% of the total transportation-related emissions [4].

Air pollution has had some seriously negative impacts on the health of residents in cities, especially in big cities in developing countries. Studies found that about 3.3 million premature deaths per year were caused by outdoor air pollution on a global scale [5], with Asia being the most affected

area [6]. Controlling greenhouse gas and pollutant emission is a major priority in protecting the global environment.

A new and typical problem is the dangerous level of air pollution caused by vehicular traffic and industrial emissions. In such an environment, travelers are exposed to higher levels of traffic-related particulate matter (PM) and higher health risks than the general population [7]. The exposure of travelers to traffic-related PM is particularly serious in developing countries. One problem is that motorcycles and scooters (hereafter, "motorcycles") are an important and essential means of transportation in many developing countries, such as Vietnam, India, China, Brazil, and Thailand. For example, in Vietnam, motorcycles are the primary travel mode because of their economic practicality. Unfortunately, travelers who use a motorcycle are directly exposed to traffic-related PM.

When travelers use a car or other vehicles, they are indirectly exposed to traffic-related PM. The degree of reduction of PM by the vehicle's air conditioning system is limited, especially for finer particles, such as PM2.5 (i.e., particles with a mean diameter of 2.5 μm or less). This is exacerbated by the fact that most people in developing countries cannot afford to buy an expensive car equipped with a powerful and efficient air conditioning system that could mitigate the problem. Even when such a system is available, many drivers travel with the window open with the idea that driving for a long time in an airtight vehicle with the vehicle's air conditioning system operating damages their health.

For these reasons, travelers in developing countries are exposed to high concentrations of traffic-related PM. The damage caused by this pollution has become a growing concern for urban citizens and the governments responsible for protecting their health.

The increasingly serious problem of air pollution caused by vehicles and traffic congestion has created enormous challenges for transportation planners, particularly since there is strong evidence that these problems affect travel modes and behavior. For example, in Vietnam's Ho Chi Minh City, most travelers wear thick masks to prevent dust and particles from entering their mouths and noses during travel.

Although time is important to most travelers, some travelers consider both the value of time (VOT) and the health risk caused by their travel when they choose routes. This means that the perception of pollution damage has become a critical factor that affects route choices. As a result, it is no longer sufficient to just consider the goal of minimizing travel times when designing urban transportation networks, and both transportation planners and travelers must assess the importance of pollution damage. This is problematic because it complicates consideration of how to solve the classical Braess paradox [8], in which efforts to increase the capacity of a network by adding new routes can instead decrease its throughput. Specifically, an analysis that only considers the VOT of travelers is unpractical because it cannot minimize pollutant emission or traveler perceptions of the health risk created by pollution [9]. To address this problem and provide a more meaningful basis for transportation planning and policy development, we propose a new transportation network model that considers both the VOT and traveler perceptions of pollution damage.

The rest of this paper is organized as follows. Section 2 presents the literature review. In Section 3, we propose a new equilibrium model that simultaneously considers the health damage and the travel time, and analyze a new transportation paradox that arises from this model. Section 4 proposes an improved model to analyze how different transport modes influence the Braess paradox. Section 5 provides a statistical analysis. We conclude with a discussion of the implications of this work in Section 6. Section 7 presents the conclusions.

## 2. Literature Review

### 2.1. Braess Paradox

The Braess paradox has been extensively studied in the context of transportation, telecommunication, mechanical, electrical, and computer networks, as well as for large-scale random graphs and large sparse graphs [10–14]. Research suggests that the paradox can be explained by the fact that individual entities act

separately when making their travel route choices and that the lack of cooperative decision-making can force the system as a whole to operate sub-optimally. Studies of the Braess paradox have attracted attention in transportation planning and operations research. Recent research has extended the paradox to more general contexts and addressed the contributions of elastic demand [15,16], network reliability [17,18], emission considerations [19,20], time-dependent considerations [21], combined distribution and assignment [22], stochastic assignment [23,24], dynamic assignment [25–27], transit assignment [28], and boundedly rational user equilibrium [29].

Previous studies have discovered that travel time could greatly affect route choice. Typically, each traveler chooses a route between the starting point and destination that offers the minimum travel time. However, under this assumption, building more roads may not enhance the system's performance (in terms of its ability to achieve the goals of its users) or improve utilization of the network's capacity (i.e., lead to the Braess paradox). Therefore, detecting the conditions under which the Braess paradox will occur is a matter of critical importance for network planners.

Researchers have attempted to identify these conditions using various link-cost or link-congestion functions. Dafermos and Nagurney [30] proposed using a positive semidefinite matrix to test whether the paradox occurs with asymmetric link travel times. The authors of [31,32] examined the necessary and sufficient conditions when the Braess paradox occurs for general networks with linear link-cost functions. Reference [33] proved that there is at least one O/D pair connected by a new path so that the Braess paradox does not (does) occur, as the proposed test matrix is positive (negative) and semi-definite. Reference [34] showed that the Braess paradox only occurs when the total travel demand falls within a certain intermediate range of demands. Recently, based on the findings of Pas and Principio [34], Reference [35] extended previous research on the Braess paradox by considering arbitrary link-volume delay functions, and also showed that the Braess paradox occurs if and only if the total travel demand lies within a certain range of values.

### 2.2. Environment and Travelers' Choices

Research on pollution has seen substantial growth in the transportation literature [36,37]. The consensus is that air pollution from ground transportation poses a significant threat in urban areas [38,39]. Most literature focuses on how transportation planners make decisions to reduce environmental pollution. For example, Reference [40] introduced two pollution permit systems for transportation networks to provide scientific support for decision-making with the goal of pollution reduction. Reference [21] identified three distinct paradoxes that could occur in congested urban transportation networks in terms of the total emissions generated. They demonstrated that the network topology, cost structure, and travel demand structure must be considered in any policy system that is intended to reduce vehicle emissions. Reference [41] investigated the impacts of route decisions on vehicle energy consumption and emission rates for different vehicle types using microscopic and macroscopic emission estimation tools. Reference [42] reviewed the literature on applications and approaches related to designing and managing road networks to explicitly address environmental concerns.

Some other studies have addressed the impact of air pollution derived from vehicle emissions on route choice. Reference [43] examines the contrast between traditional travel cost factors and personal exposure to PM10 in optimum route choice selections. Reference [44] developed a new method for incorporating the estimated inhaled mass of PM2.5 into walking route calculations; with their method, a low air pollution inhalation route can be found. Reference [45] found that an appropriate choice of route through an urban area might significantly reduce the air pollution exposure, and a web-based route planner for selecting the low exposure route through the city could be good for the public. Reference [46] proposed a healthier route planning (HRP) method to minimize personal travel exposure risk to air pollution by integrating techniques of fine-scale mapping of air pollutant concentration, risk weight estimation of road segment exposure to air pollutants, and the dynamic Dijkstra algorithm.

*2.3. Summary and Research Problem*

In the existing literature, some in-depth studies on the Braess paradox have been conducted, and scholars have also found that air pollution would affect travelers' route choices. However, how does air pollution derived from vehicle emissions impact the Braess paradox? The previous studies lack relevant discussions. This is also the research gap to be filled in this paper.

To fill this gap and provide more realistic solutions for the Braess paradox, the present study explores the effects of health damage on route choices and policy decisions.

Travelers often live in cities with serious air pollution and high traffic congestion. We attempted to simultaneously evaluate the effects of the health risk caused by pollution and the travel time within an urban transportation network. To do so, we used a novel method of computing a total comprehensive cost that combines the costs of above two factors. Furthermore, we predicted the conditions under which the paradox will occur and the conditions for improving system performance when both factors are considered. In particular, the impacts of different perceptions of the health risk caused by exposure to pollution on the occurrence of the paradox are analyzed. The whole research framework is shown in Figure 1.

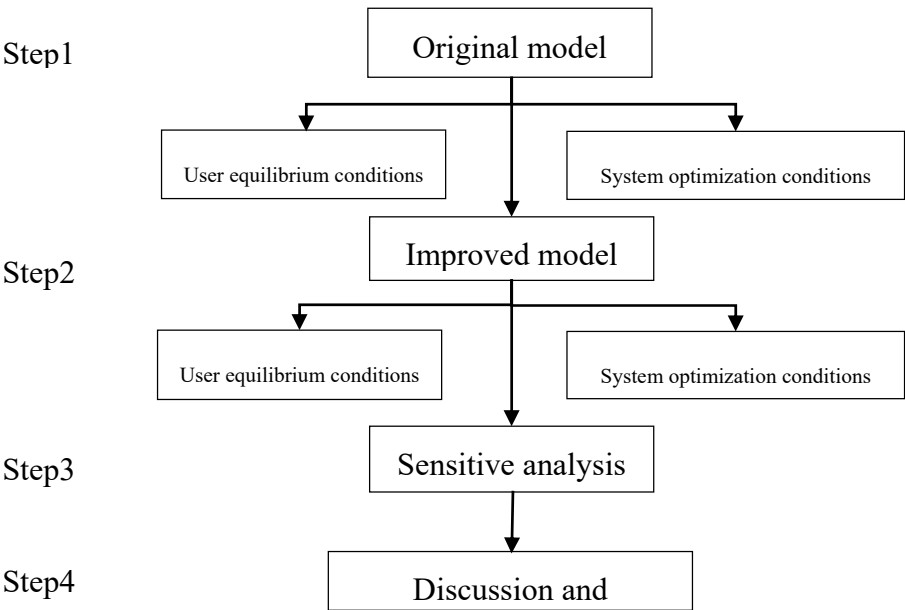

**Figure 1.** Research framework.

## 3. Problem Definition and Formulation

In this section, we propose a new equilibrium model for urban road selection in the context of a Braess network, and analyze some kinds of situations that lead to the Braess paradox.

*3.1. Problem Definition*

We used the same network configuration defined by [8] in our analysis (Figure 2). Figure 2a depicts the original network, and Figure 2b depicts the revised network with an added link (pq). Travelers can take two routes (opd and oqd) or three routes (opd, oqd, opqd) from the origin (o) to the destination (d) in the original and revised networks, respectively. For convenience, we refer to routes opd, oqd, and opqd as routes 1, 2, and 3, respectively.

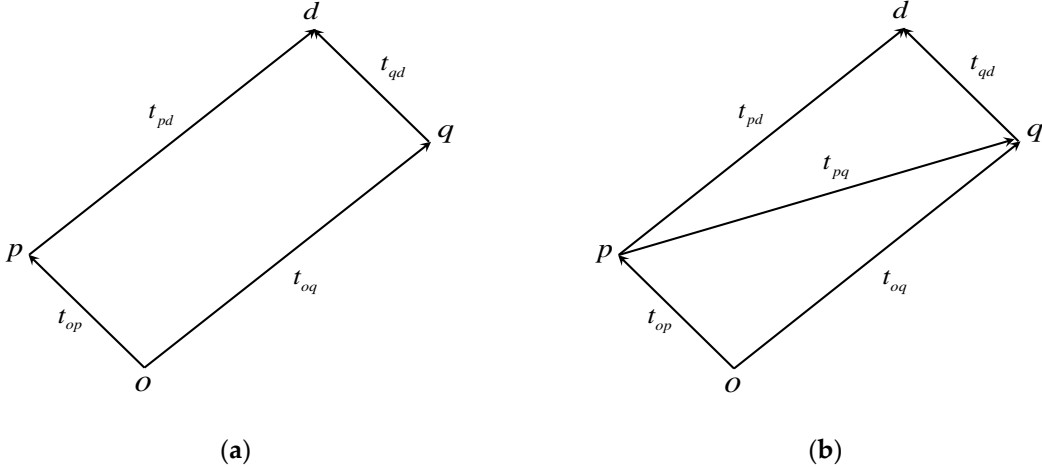

**Figure 2.** Two networks that offer different routes between the origin (o) and destination (d) of the traveler (source: [8]). (**a**) The original network with four links. (**b**) The revised network with five links. The t values represent travel times along each link.

To analyze the travel time and health risk perceived by travelers, the traveler flows are equivalently transformed into traffic flows (e.g., flows of cars and motorcycles), and each traveler corresponds to one element of the traffic flow. Typically, it is assumed that the travel time (*t*) is a function of the traffic flow and the free-flow travel time. In this paper, the travel time $t_{ij}$ along link ij is calculated using the following equation [14,22]:

$$t_{ij} = \alpha_{ij} + \beta_{ij} \cdot x_{ij}, \tag{1}$$

where $\alpha_{ij}$ denotes the free-flow travel time along link *ij*, $\beta_{ij}$ denotes the congestion time (or delay parameter) per unit of traffic flow along link *ij*, and $x_{ij}$ denotes the total traffic flow along link *ij*. In addition, we have defined links op and qd as bottleneck-type links because they are assumed to be so short that their free-flow travel time is zero. Based on the symmetry consideration addressed by [8], the relevant parameters are set as follows: $\alpha_{oq} = \alpha_{pd} = \alpha_1$, $\alpha_{pq} = \alpha_2$, $\beta_{op} = \beta_{qd} = \beta_1$, and $\beta_{oq} = \beta_{pd} = \beta_{pq} = \beta_2$. In addition, let $\beta_1 > \beta_2$, which represents a situation in which the congestion level along the bottleneck-type links is worse than that on the other links. Let set $t_i$ mean the travel time of route *i*, where $t_{ij} \in t_i$, and *j* means the sub-route of route *i*.

Based on the above definitions, the travel time along each of the three routes can be calculated as follows:

$$t_1 = \sum t_{1j} = \alpha_1 + \beta_1 \cdot x_{op} + \beta_2 \cdot x_{pd} \tag{2}$$

$$t_2 = \sum t_{2j} = \alpha_1 + \beta_1 \cdot x_{qd} + \beta_2 \cdot x_{oq} \tag{3}$$

$$t_3 = \sum t_{3j} = \alpha_2 + \beta_1 \cdot x_{op} + \beta_1 \cdot x_{qd} + \beta_2 \cdot x_{pq}, \tag{4}$$

where $t_1$, $t_2$, and $t_3$ represent the travel times along routes 1, 2, and 3, respectively.

The health damage of travelers depends mainly on their exposure to pollution, their degree of concern about potential health damage, and the degree of pollution. The highest PM fraction concentrations were observed during the congestion period. Typically, the exposure level of travelers taking a car will be lower than that of travelers taking a motorcycle. However, as we mentioned before, this is not correct for PM2.5 because most cars in developing countries have an inexpensive air-purification system that cannot cope with such small particles. Reference [47] found that even after taking the increased respiration rate of cyclists into consideration, car drivers seem to be more exposed to airborne pollution than cyclists. Reference [48] showed that average PM concentrations were 3.3 times higher inside auto-rickshaws than the ambient level. In addition, motorcycles are the main means of transportation in such countries, and most travelers are, therefore, completely exposed

to traffic-related PM. Therefore, it is reasonable for us to account for perceptions of the health risk of pollution in the context of developing countries.

In addition, groups differ in their degree of concern about health damage. A high degree of concern may result in high health damage.

A certain amount of emission is created when a vehicle traverses link *ij*. The degree of air pollution encountered along link *ij* depends on many factors, such as the number of vehicles, speed of the vehicles, vehicle types, and fuel consumption [40]. For example, elevated PM levels in cars were related to locations with high traffic volumes and high emissions from the internal combustion engines of other vehicles [49]. Reference [50] pointed out that occupational exposure to urban-traffic-related air pollutants led to a significant induction of cytogenetic damage in peripheral lymphocytes of traffic policemen and taxi drivers. Comparing to gasoline vehicles under the same conditions of traffic congestion, electric vehicles have less emissions and within-vehicle air pollution exposure is better, so the damage to human health is relatively lower [51].

Given that we aimed to analyze how the perception of the health risk of pollution affects route choice, we initially assumed that all travelers have the same level of exposure to air pollution and the same degree of concern about health damage. As a result, we can simplify the problem by assuming that the degree of air pollution along link *ij* is only related to traffic flows and travel time along this link. Based on these assumptions, the health damage along link *ij* ($e_{ij}$) can be defined as follows:

$$e_{ij} = h \cdot \vec{x_{ij}} \cdot \vec{t_{ij}}, \tag{5}$$

where $h$ represents the perception factor, which reflects the exposure to air pollution and the concern about health damage, and $x_{ij}$ represents the traffic flows along link *ij*, which determines the magnitude of the air pollution. Then, the health damage of travelers who take each of the three routes can be defined as follows:

$$e_1 = h \cdot \left( \vec{x_{op}}, \vec{x_{pd}} \right) \cdot \vec{t_{1j}} \tag{6}$$

$$e_2 = h \cdot \left( \vec{x_{oq}}, \vec{x_{qd}} \right) \cdot \vec{t_{2j}} \tag{7}$$

$$e_3 = h \cdot \left( \vec{x_{op}}, \vec{x_{pq}}, \vec{x_{qd}} \right) \cdot \vec{t_{3j}}, \tag{8}$$

where $e_1$, $e_2$, and $e_3$ represent the health damage along routes 1, 2, and 3, respectively.

To determine how the travel time and health damage affect route choice, the travel time and the health damage are converted into costs by introducing a corresponding cost conversion factor. Then, the comprehensive cost along link *ij* for a traveler ($C_{ij}$) can be calculated as follows:

$$C_{ij} = \theta_1 \cdot t_{ij} + \theta_2 \cdot e_{ij}, \tag{9}$$

where $\theta_1$ denotes the VOT, which represents the opportunity cost for the time that a traveler spends on their journey, and $\theta_2$ denotes the value of health damage, which represents the opportunity cost for health damage caused by the air pollution that a traveler encounters while traveling along a link.

The comprehensive costs of traveling along routes 1, 2, and 3 for a traveler ($C_1$, $C_2$, and $C_3$, respectively) can be calculated as follows:

$$C_1 = \theta_1 \cdot (t_1 + \delta \cdot e_1) \tag{10}$$

$$C_2 = \theta_1 \cdot (t_2 + \delta \cdot e_2) \tag{11}$$

$$C_3 = \theta_1 \cdot (t_3 + \delta \cdot e_3). \tag{12}$$

Furthermore, let Q be the total number of travelers who flow from the origin to the destination, and let $f_1$, $f_2$, and $f_3$ to be the numbers of travelers that choose routes 1, 2, and 3, respectively. Thus, $Q = f_1 + f_2 + f_3$, where $f_3 = 0$ in the original network, as shown in Figure 2a. Thus, the total

comprehensive cost of using the original and revised networks for all travelers ($C_0$ and $C_r$, respectively) can be calculated as follows:

$$C_o = f_1 \cdot C_1 + f_2 \cdot C_2 \tag{13}$$

$$C_r = f_1 \cdot C_1 + f_2 \cdot C_2 + f_3 \cdot C_3. \tag{14}$$

### 3.2. User Equilibrium Conditions

Having determined the total comprehensive cost for the travelers, we will now focus on the possibility that a traffic flow paradox (i.e., a Braess paradox) will occur. By definition, economically rational individuals will always choose the routes that will minimize their travel cost. In this paper, the traveler's comprehensive cost will replace the single travel time cost addressed by the traditional user equilibrium model. That is, a user equilibrium (UE) can be reached when no traveler can decrease their comprehensive cost by unilaterally shifting to another route, and at that point, the flow distribution within the transportation network attains a steady state (i.e., a user equilibrium exists).

The UE condition for the four-link network in Figure 2a can be presented as follows [52]:

$$C_1 = C_2. \tag{15}$$

The traffic flows along routes 1 and 2 can then be described as follows:

$$f_1 = f_2 = \frac{1}{2}Q. \tag{16}$$

The total comprehensive cost of the four-link network is then calculated as follows:

$$C_o = \frac{\theta_1}{2}Q(\alpha_1 + (\beta_1 + \beta_2)Q) + \frac{\theta_2}{4}hQ^2((\beta_1 + \beta_2)Q + 2\alpha_1). \tag{17}$$

Similarly, the UE condition, flow distribution, and total comprehensive cost for the five-link network in Figure 1 can be described as follows:

$$C_1 = C_2 = C_3 \tag{18}$$

$$f_1 + f_2 + f_3 = Q \tag{19}$$

$$f_1 = f_2. \tag{20}$$

The simultaneous solution of the above equations can be obtained (negative roots are deleted in all following solutions):

$$f_3 = \frac{-b_1 + \sqrt{b_1{}^2 - 4a_1c_1}}{2a_1}, \tag{21}$$

where $a_1 = \frac{1}{4}\theta_2 h(\beta_1 + 3\beta_2)$, $b_1 = \frac{1}{2}\theta_1(\beta_1 + 3\beta_2) + \frac{1}{2}\theta_2 h((\beta_1 + \beta_2)Q + 3\alpha_1)$, $c_1 = \frac{1}{4}\theta_2 h(\beta_1 - \beta_2)Q^2 + \frac{1}{2}(\theta_1(\beta_1 - \beta_2) - \theta_2 h\alpha_1)Q + \theta_1(\alpha_2 - \alpha_1)$.

To expressly test whether each road in the five-link network is being used, we performed the following analyses:

(1)  If $Q_1 > Q_2$, we obtain $f_1 = f_2 = 0$ and $f_3 = Q$.

$$Q_1 = \frac{-b_2 + \sqrt{b_2{}^2 - 4a_2c_2}}{2a_2}, \tag{22}$$

where, $a_2 = \theta_2 h(\beta_1 + \beta_2)$, $b_2 = \theta_1(\beta_1 + \beta_2) + \theta_2 h\alpha_2$, $c_2 = \theta_1(\alpha_2 - \alpha_1)$.

(2)   If $Q_1 < Q_2$, we have $f_1 = f_2 = Q/2$ and $f_3 = 0$,

$$Q_2 = \frac{-b_3 + \sqrt{b_3{}^2 - 4a_3c_2}}{2a_3} \tag{23}$$

where $a_3 = \frac{\theta_2 h(\beta_1 - \beta_2)}{4}$, $b_3 = \frac{1}{2}(\theta_1(\beta_1 - \beta_2) - \theta_2 h\alpha_1)$.

(3)   If $Q \in (Q_1, Q_2)$, $f_3 = \frac{-b_1 + \sqrt{b_1{}^2 - 4a_1c_1}}{2a_1}$, $f_1 = \frac{Q - f_3}{2}$.

### 3.3. System Optimization Conditions

The addition of a new link can lower the system's performance if the total travel demand lies within a certain interval. Nevertheless, the system's performance can be improved through the use of optimization methods. Specifically, the performance can be improved if we can look at this problem from a global perspective rather than assuming that each traveler acts independently, as is the case in a traditional Braess paradox. Thus, we will also analyze the comprehensive cost under improved system optimization (SO) conditions instead of based solely on the travel time cost, as is the case under the traditional system optimization conditions.

To support the application of the SO conditions in our analysis, we have introduced a marginal comprehensive cost (MCC), which represents the marginal contribution to the comprehensive cost of a certain link. The principle of the calculation for MCC can be illustrated by using the example of the MCC for route 1. First, the marginal cost of travel time (MT) along link op is calculated as follows:

$$MT_{op} = \frac{d\beta_1 x_{op}^2}{dx_{op}}. \tag{24}$$

Similarly, the MT along link pd is determined as follows:

$$MT_{pd} = \frac{d(\alpha_1 + \beta_2 x_{pd})x_{pd}}{dx_{pd}}. \tag{25}$$

Then, the MT along route 1 is calculated as follows:

$$MT_1 = \frac{d\beta_1 x_{op}^2}{dx_{op}} + \frac{d(\alpha_1 + \beta_2 x_{pd})x_{pd}}{dx_{pd}}. \tag{26}$$

In the same way, we can calculate the marginal cost of the health damage (ME) along route 1 as follows:

$$ME_1 = \frac{dhx_{op} \cdot \beta_1 x_{op} \cdot x_{op}}{dx_{op}} + \frac{dhx_{pd} \cdot (\alpha_1 + \beta_2 x_{pd}) \cdot x_{pd}}{dx_{pd}}. \tag{27}$$

Thus, the MCC is the sum of the two marginal costs, multiplied by their respective conversion factors ($\theta_1$ and $\theta_2$). That is, the MCC of each traveler along routes 1, 2, and 3 (MCC1, MCC2, and MCC3, respectively) can be calculated as follows:

$$MCC_1 = \theta_1(\alpha_1 + 2\beta_1 x_{op} + 2\beta_2 x_{pd}) + \theta_2 h(3\beta_1 x_{op}{}^2 + 2\alpha_1 x_{pd} + 3\beta_2 x_{pd}{}^2) \tag{28}$$

$$MCC_2 = \theta_1(\alpha_1 + 2\beta_2 x_{oq} + 2\beta_2 x_{qd}) + \theta_2 h(3\beta_1 x_{oq}{}^2 + 2\alpha_1 x_{qd} + 3\beta_2 x_{qd}{}^2) \tag{29}$$

$$MCC_3 = \theta_1(\alpha_2 + 2\beta_1 x_{op} + 2\beta_2 x_{pq} + 2\beta_1 x_{qd}) + \theta_2 h(3\beta_1(x_{op}{}^2 + x_{qd}{}^2) + 2\alpha_2 x_{pq} + 3\beta_2 x_{pq}{}^2). \tag{30}$$

If we let $MCC_1 = MCC_2 = MCC_3$ [52], then the flow distribution under the SO conditions can be calculated as follows:

(1)　As $Q \in [0, Q_3]$, $f_3 = Q_3$

$$Q_3 = \frac{-b_3 + \sqrt{b_3{}^2 - 4a_3c_2}}{2a_3},\tag{31}$$

where $a_3 = 3\theta_2 h(\beta_2 + \beta_1)$, $b_3 = 2(\theta_1(\beta_1 + \beta_2) + \theta_2 h\alpha_2)$.

(2)　$Q \in [Q_3, Q_4]$, $f_3 = \frac{-b_4 + \sqrt{b_4{}^2 - 4a_4c_3}}{2a_4}$

where $a_4 = \frac{3}{4}\theta_2 h(\beta_1 + 3\beta_2)$, $b_4 = \theta_1(3\beta_2 + \beta_1) + \theta_2 h(\frac{3}{2}(\beta_1 + \beta_2)Q + 2\alpha_2 + \alpha_1)$, $c_4 = \frac{3}{4}\theta_2 h(\beta_1 - \beta_2)Q^2 + (\theta_1(\beta_1 - \beta_2) - \theta_2 h\alpha_1)Q + \theta_1(\alpha_2 - \alpha_1)$.

$$Q_4 = \frac{-b_5 + \sqrt{b_5{}^2 - 4a_5c_2}}{2a_5},\tag{32}$$

where $\alpha_5 = \frac{3}{4}(\beta_1 - \beta_2)$, $b_5 = \theta_1(\beta_1 - \beta_2) - \theta_2 h\alpha_1$.

(3)　$Q \in [Q_4, +\infty]$, $f_3 = 0$.

## 4. Improved Model

In the urban transportation network, to address the issue of traffic jams, the government can choose different transport modes, such as ordinary roads, quick-pass roads, and subways. There are obvious differences in emissions, traffic speed, and other aspects among these different modes of transportation, which will impact the traffic flows' distribution and paradox. So, these sections will consider the impact of new roads using different modes of transportation on the traffic paradox.

It is assumed that the health damage coefficient of the original road is $h_1$, and the health damage coefficient of the new traffic mode is $h_2$. Obviously, as $h_1 = h_2$, the improved model becomes the original model.

Let $h = \{h_1, h_2\}$; we get that

$t_1 = \alpha_1 + \beta_1 x_{op} + \beta_2 x_{qd}$

$e_1' = h_1 \cdot \left(\vec{x_{op}}, \vec{x_{pd}}\right) \cdot \vec{t_{1j}}$

$C_1 = \theta_1 t_1 + \theta_2 e_1'$

$t_3 = \alpha_1 + \beta_1(x_{op} + x_{qd}) + \beta_2 x_{pq}$

$e_3' = \vec{h} \cdot \left(\vec{x_{op}}, \vec{x_{pd}}\right) \cdot \vec{t_{3j}}$

$C_3 = \theta_1 t_3 + \theta_2 e_3'$

### 4.1. UE Condition

Under the UE condition, $C_1 = C_2 = C_3$, we get that

(1)　$Q > Q_5$, $f_3 = 0$

$$Q_5 = \frac{-b_5 + \sqrt{b_5{}^2 - 4a_5c_2}}{2a_5},\tag{33}$$

where $a_5 = \frac{\theta_2 h_1(\beta_1 - \beta_2)}{4}$, $b_5 = \frac{1}{2}(\theta_1(\beta_1 - \beta_2) - \theta_2 h_1 \alpha_1)$

(2)　$Q < Q_6$, $f_3 = Q$

$$Q_6 = \frac{-b_7 + \sqrt{b_7{}^2 - 4a_7c_5}}{2a_7},\tag{34}$$

where $a_7 = \theta_2(h_1\beta_1 + h_2\beta_2)$, $b_7 = \theta_1(\beta_1 + \beta_2) + \theta_2 h_2 \alpha_2$

(3)　$Q_6 < Q < Q_5$, $f_3 = \frac{-b_6 + \sqrt{b_6{}^2 - 4a_6c_6}}{2a_6}$

where $a_6 = \frac{\theta_2 h_1(\beta_1 - \beta_2)}{4} + \theta_2 h_2\beta_2$, $b_6 = \frac{1}{2}\theta_1(\beta_1 + 3\beta_2) + \theta_2(h_1(\beta_1 + \beta_2)Q + \frac{1}{2}\alpha_1(h_1 + 2h_2))$, $c_6 = \frac{1}{4}\theta_2 h_1(\beta_1 - \beta_2)Q^2 + \frac{1}{2}(\theta_1(\beta_1 - \beta_2) - \theta_2 h_1\alpha_1)Q + \theta_1(\alpha_2 - \alpha_1)$.

### 4.2. SO Condition

Under the SO condition,$MCC_1 = MCC_2 = MCC_1$
$MCC_1 = \theta_1(2\beta_1 x_{op} + \alpha_1 + 2\beta_2 x_{pd}) + \theta_2 h_1(3\beta_1 x_{op}{}^2 + 2\alpha_1 x_{pd} + 3\beta_2 x_{pd}{}^2)$
$MCC_3 = \theta_1(2\beta_1 x_{op} + \alpha_2 + 2\beta_2 x_{pq} + 2\beta_1 x_{qd}) + \theta_2 h_1(3\beta_1(x_{op}{}^2 + x_{qd}{}^2)) + \theta_2 h_2(2\alpha_2 x_{pq} + 3\beta_2 x_{pq}{}^2)$.
We get that

(1)  $Q \in [0, Q_7]$, $f_3 = Q$

$$Q_7 = \frac{-b_7 + \sqrt{b_7{}^2 - 4a_7 c_2}}{2a_7}, \tag{35}$$

where $a_7 = 3\theta_2(h_2\beta_2 + h_1\beta_1)$, $b_7 = 2(\theta_1(\beta_1 + \beta_2) + \theta_2 h_2 \alpha_2)$

(2)  $Q \in [Q_7, Q_8]$, $f_3 = \frac{-b_8 + \sqrt{b_8{}^2 - 4a_8 c_8}}{2a_8}$

where $a_8 = 3\theta_2 h_2 \beta_2 + \frac{3}{4}\theta_2 h_1(\beta_1 - \beta_2)$, $b_8 = \theta_1(3\beta_2 + \beta_1) + \theta_2(\frac{3}{2}h_1(\beta_1 + \beta_2)Q + 2h_2\alpha_2 + h_1\alpha_1)$,
$c_8 = \frac{3}{4}\theta_2 h_1(\beta_1 - \beta_2)Q^2 + (\theta_1(\beta_1 - \beta_2) - \theta_2 h_1 \alpha_1)Q + \theta_1(\alpha_2 - \alpha_1)$

$$Q_8 = \frac{-b_9 + \sqrt{b_9{}^2 - 4a_9 c_2}}{2a_9}, \tag{36}$$

where $\alpha_9 = \frac{3}{4}(\beta_1 - \beta_2)$, $b_9 = \theta_1(\beta_1 - \beta_2) - \theta_2 h_1 \alpha_1$

(3)  $Q \in [Q_8, +\infty]$, $f_3 = 0$.

According to the improved model, we can deduce the following three propositions.

**Proposition 1:** *The flows of route 3 under the SO condition are smaller than those under the UE condition.*

**Proof:** $\because a_8 = 3a_6$
$b_8 < 3b_6$
$c_8 < 3c_6$.
So, $f_3^{SO} < \frac{-3b_6 + \sqrt{(3b_6)^2 - 4(3a_6)(3c_6)}}{2(3a_6)} = f_3^{UE}$.  $\square$

This means that under UE condition, there are more traffic flows coming into the new road than under the SO condition. The choice under the SO condition is somewhat more rational.

**Proposition 2:** *There is a non-paradox interval under the SO condition.*

**Proof:** It can be known that
$MCC_1 > 0$, $MCC_3 > 0$, which means that all marginal costs of route 1, route 2, and route 3 are increased.  $\square$

So, under the SO condition, at the flow point $Q = Q_8$, $C_r(f_3 = 0) = C_0$, and $MCC_3(f_3 = 0) = MCC_1(f_3 = 0)$. If we take some flows of $f_3$ to $f_1$, the marginal cost $MCC_3 < MCC_1$, and the total cost will be increased.
Then, we can get Proposition 2.

**Proposition 3:** *If $\theta_1(\alpha_2 - \alpha_1) + \theta_1(\beta_1 - \beta_2)Q_5 + \theta_2 h_1(\frac{3}{4}\beta_1 Q_5^2 - \beta_2 Q_5^2 - \alpha_1 Q_5) > 0$, there is a non-paradox interval under the UE condition.*

**Proof:** Under the UE condition, at the flow point $Q = Q_5$, $C_r(f_3 = 0) = C_0$.
If $MCC_3 > MCC_1$, there is a Pareto improving space when $Q < Q_5$.
So, there is a sufficient condition for a non-paradox interval under the UE condition, which is $MCC_3 > MCC_1$ at the flow point $Q = Q_5$.  $\square$

That is, $\theta_1(\alpha_2 - \alpha_1) + \theta_1(\beta_1 - \beta_2)Q_5 + \theta_2 h_1\left(\frac{3}{4}\beta_1 Q_5^2 - \beta_2 Q_5^2 - \alpha_1 Q_5\right) > 0$.

## 5. Numerical Experiments

### 5.1. Network Flows and System Performance under UE and SO Conditions

The relevant scenarios for our proposed model involve large cities in developing countries that have a large number of motorcycles, such as China's Chongqing City, in which the number of motorcycles is more than 1.83 million, and it ranked first in China in 2019. The annual report on traffic development in Chongqing's metropolitan area suggests that vehicle ownership in Chongqing ranked first in China, reaching $6 \times 10^6$ in 2019, with motorcycles, cars and buses, trucks, and other vehicles (e.g., tractors and trailers) accounting for 30.1%, 60.7%, 8.6%, and 0.6% of the total vehicle ownership, respectively. In addition, there is severe air pollution in Chongqing. Based on the data released by the Ministry of Environmental Protection of China (http://106.37.208.228:8082/), the annual mean concentrations of PM2.5 and PM10 in the Chongqing metropolitan area in 2019 were 38 and 60 μg/m$^3$, respectively. PM2.5 exceeds the Grade II standards in the Chinese Ambient Air Quality Standards (35 μg/m$^3$ for PM2.5). Moreover, the number of days on which any pollutant concentration exceeded Grade II standards totaled 49 days annually for a non-attainment rate of 13.4%, based on the most recent version of these standards (GB3095-2012). PM2.5 was the largest contributor to air pollution in Chongqing City based on the number of non-attainment days. Therefore, travelers in Chongqing tend to wear thick masks to reduce pollution damage (Figure 3). Thus, they appear to consider both the VOT and pollution damage when choosing their route.

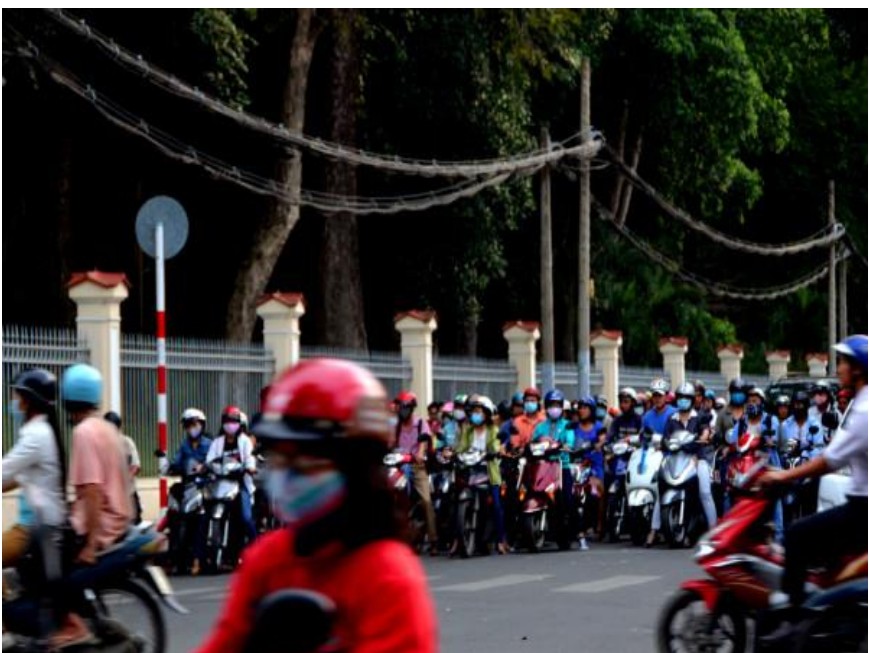

**Figure 3.** Motorcycle drivers wearing thick masks.

The parameter values used in the present experiment are listed in Table 1, which synthesizes the congestion function parameters from Braess (1968) [8] and the pollution factors from Nagurney (2000) [40].

**Table 1.** Parameter values used in the current experiment.

| Parameter | $\alpha_1$ | $\alpha_2$ | $\beta_1$ | $\beta_2$ | $\theta_1$ | $\theta_2$ | $h_2$ |
|---|---|---|---|---|---|---|---|
| Value | 10 | 1.5 | 2 | 1 | 1 | 3 | 0.1 |

In order to find out whether the Braess paradox is happening, the paper uses $C_r$-$C_0$ as the ordinate. When the value of the ordinate is less than 0, it means that there is no Braess paradox; otherwise, the paradox will occur. The results are shown in Figure 4.

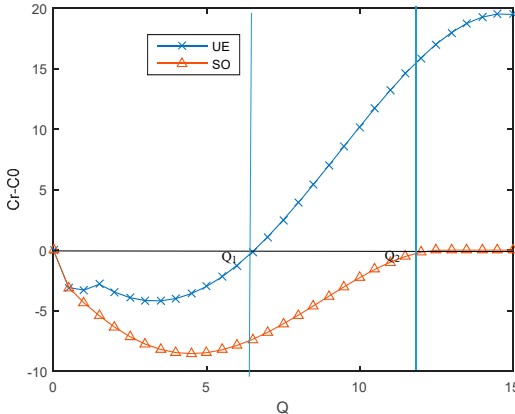

**Figure 4.** The performance under the user equilibrium (UE) and system optimization (SO) conditions.

(1) Braess paradox traffic flow interval

The total cost under the SO condition is lower than the total cost under the UE condition, which means that the overall equilibrium condition is better than the local equilibrium condition. With the change of $Q$, under the UE condition, $[0, Q_1]$ is a non-paradox interval, and under the SO condition, $[0, Q_2]$ is a non-paradox interval. The non-paradox interval under the SO condition is larger than that under the UE condition, which has more space for traffic improvement.

When $Q$ increases to break through the critical threshold, it will enter into the paradox interval, which means that if a new road does not occur in the Braess paradox, the whole traffic flow must be controlled. Only when the overall traffic flow is less than the critical value will there be a non-paradox interval. When the traffic flow of an area exceeds the critical value, the traffic paradox will always appear.

Figure 5 shows the influence of the total traffic flow ($Q$) on the new road flow ($f_3$) under two different conditions. It can be seen that under the UE conditions, more traffic will enter the new road, and the corresponding total cost is relatively high. This means that most of the traffic entering the new road under the UE condition is not a rational choice. Under the condition of asymmetric information, in the equilibrium game of traffic participants, the prisoner's dilemma is difficult to avoid, and the irrational choice to enter the new road will increase the overall cost.

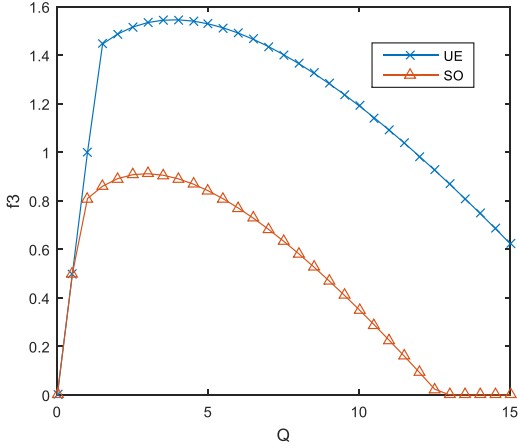

**Figure 5.** The impact of traffic flow ($Q$) on new road flow ($f_3$).

(2) The impacts of health damage of the new road

Figure 6 reflects the impact of the health damage coefficient on performance improvement. As $Q$ is fixed ($Q = 1.5$ in the example), with the increase of the new road health damage coefficient (h2), the total cost of the transportation network under both conditions will increase, while the total cost under UE is always higher than that under the SO condition. On the one hand, this shows that the health damage caused by the new road will have a positive impact on the comprehensive cost of the whole traffic network. On the other hand, it also shows that SO is a better traffic equilibrium condition than UE.

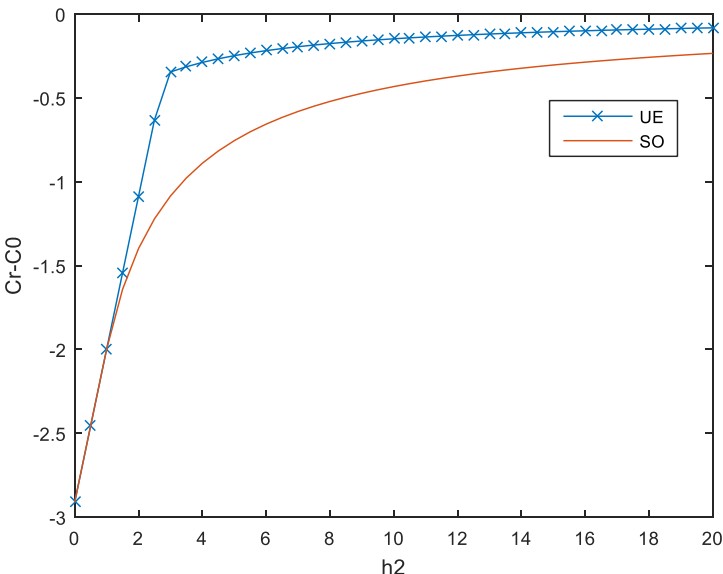

**Figure 6.** The impacts of health damage of the new road on the performance improvement.

In a word, in the traffic network planning, it is necessary to control the trend of traffic flow from the network as a whole instead of letting the traffic participants choose by themselves. The individual optimal choice cannot achieve the overall optimization. At this time, we need not only the full transparency of information, but also guidance of the traffic flow. When the traffic flow is too large and exceeds the critical threshold, it is not advisable to build the new route when the whole flow falls into intervals $[Q_2, +]$ of Figure 4, since there are no traffic flows along the new route, and the construction of a new link will waste both time and money and will increase pollution.

Moreover, as highlighted by Pas and Principio (1997), whether or not the Braess paradox occurs depends on the problem's parameter definitions. The bounds of the range in which the new paradox occurs depend on the values of the set of parameters for the functions that describe the health damage and the link travel time. The paradoxes operate similarly with and without consideration of the health damage.

## 5.2. Effects of Different Categories of Pollution Susceptibility

To analyze the effects of different health damages on the system's performance and route choice, we tested the effects of changing different parameters. Firstly, we consider the effects of two key variables, which are $Q$ and $h_2$, on the total cost of the whole traffic network. The impacts of $Q$ and $h_2$ on the total cost are shown in Figure 7. The dark-color curved surface is $C_r$-$C_0$ under the UE condition, and the white curved surface is $C_r$-$C_0$ under the SO condition.

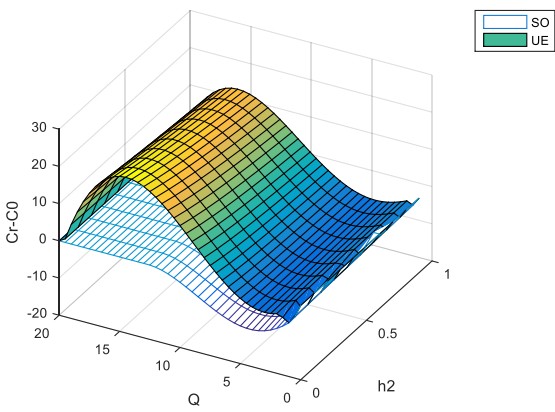

**Figure 7.** The impacts of $Q$ and $h_2$ on the total cost.

When the ordinate value $C_r$-$C_0$ is less than 0, it means that there is no paradox interval. On the contrary, when $C_rC_0$ is bigger than 0, the paradox appears. The dark UE surface is always higher than the SO surface, which means that the total network cost under the SO condition is less than that under the UE condition. The SO condition is the global equilibrium condition, while the UE condition is the individual equilibrium condition. The global optimal selection is better than the individual optimal selection.

To further analyze the influence of different parameters on the paradox, we will adjust the values of parameters to analyze the validity of the paradox in different situations.

### 5.2.1. Sensitivity of Parameter $h_2$

It can be seen from Figure 8 that:

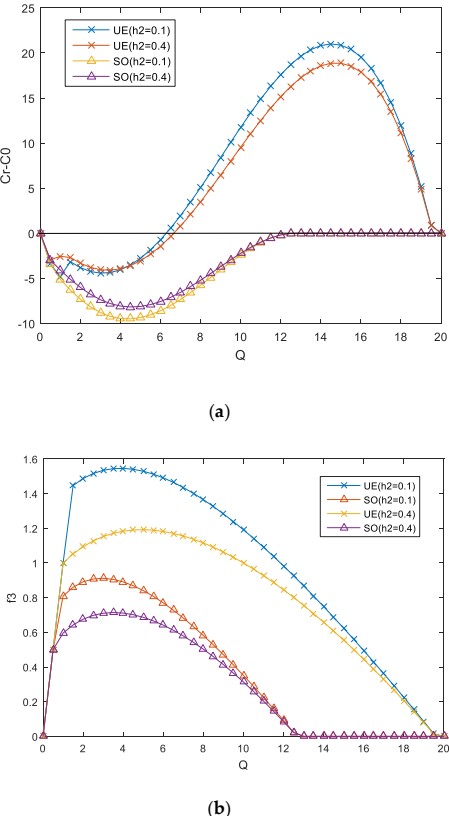

(a)

(b)

**Figure 8.** Sensitivity of parameter $h_2$: (**a**) The effects of $h_2$ on $C_r$-$C_0$; (**b**) The effects of $h_2$ on $f_3$.

(1) There is an interval where the Braess paradox is overcome in both conditions. With the increase of $h_2$, the comprehensive transportation cost will increase in different degrees under the two conditions. In practice, the emissions of different vehicles and road vary greatly, and the level of damage to human health is also significantly different. For example, due to the use of electric power, better sealing of the carriage, less pollutant emissions, and lower level of pollutants entering the carriage, by taking the subway, the harm to human health will be relatively small. On the contrary, gasoline vehicles, especially motorcycles, emit more pollutants and have poor sealing performance, thus causing higher damage to human health. When new roads are built in cities, to reduce the comprehensive transportation cost, more attention should be paid to the use of transport modes with less damage to human health, especially rail transit and other modes with large traffic volume and low emissions.

(2) The range of the non-paradox interval is also affected by $h_2$. When $h_2$ increases, the non-paradox interval increases under the UE condition, but it decreases under the SO condition. The reason for this phenomenon is that under the condition of UE, the distribution of traffic flow on different roads is determined by the individual optimal choice, which is often not a rational choice. This trend can also be seen in Figure 8b on the right. When $h_2$ decreases, much more traffic will enter the new road under the UE condition, and the excessive traffic inflow will increase the cost of congestion and health damage, which makes the paradox happen more easily.

(3) When $h_2$ increases, $f_3$ presents a downward trend, which means that the greater the damage to human health caused by new roads, the more difficult it is to attract traffic flow, and the overall performance of the traffic network becomes worse. This also fully demonstrates the value of clean traffic modes for the environment and health.

### 5.2.2. Sensitivity of Parameter $\theta$

$\theta_1$ and $\theta_2$ represent the weights of the VOT and health damage in the traffic process, respectively. When the two coefficients increase or decrease in the same proportion, the results will only change in dimension, but will not have a substantial impact. Therefore, this paper only considers the influence of $\theta_2$ on the results. $\theta_2$ can be regarded as people's attention to the health damage caused by the traffic process. As shown in Figure 9, when $\theta_2$ increases, the non-paradox interval increases from $(0, Q_1)$ to $(0, Q_2)$ under the UE condition and from $(0, Q_3)$ to $(0, Q_4)$ under the SO condition. This means that when people pay more attention to health damage, the Braess paradox is less likely to occur.

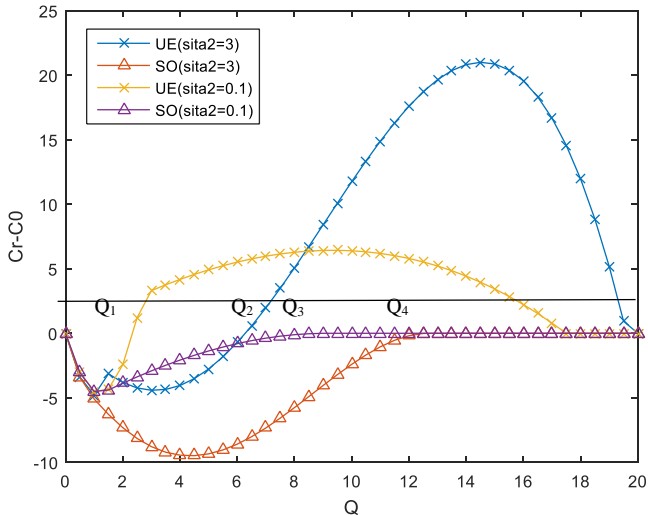

**Figure 9.** Sensitivity of parameter $\theta$.

### 5.2.3. Sensitivity of Parameter $\alpha$

Since $\alpha$ represents the road commuting time without traffic flow, it can be understood as the standard commuting time of the road. Standard commuting time depends on factors such as mode of transportation (e.g., expressways and ordinary roads), road length, etc.

As shown in Figure 10, when $\alpha_1$ is reduced, it means that the standard commuting time of the original road is reduced. The performance is reduced with $\alpha_1$ being reduced. It is easier for the paradox to appear, and even the non-paradox interval disappears under the UE condition. When the existing road is already a high-speed road, if the new road cannot effectively reduce the standard commuting time, it will be more prone to the traffic paradox.

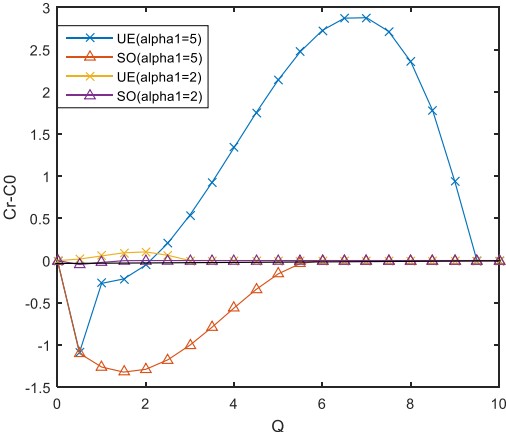

**Figure 10.** Sensitivity of parameter $\alpha_1$.

It can be seen from Figure 11a that when $\alpha_2$ increases, the operation cost of the transportation network increases under the SO condition. Interestingly, the non-paradox interval under the SO condition decreases from $(0, Q_4)$ to $(0, Q_3)$, while the non-paradox interval increases from $(0, Q_1)$ to $(0, Q_2)$ under the UE condition. This means that under the SO condition, when the standard commuting time of new roads is reduced, such as by using high-speed roads and rail transit, this can effectively reduce and overcome the traffic paradox and reduce the total traffic cost, while the UE condition is more likely to produce the paradox. The reason why it is easier to produce the paradox under the UE condition can be seen from Figure 11b on the right.

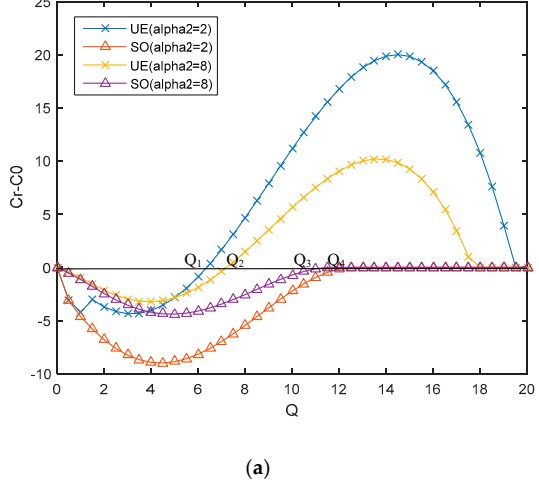

(**a**)

**Figure 11.** *Cont.*

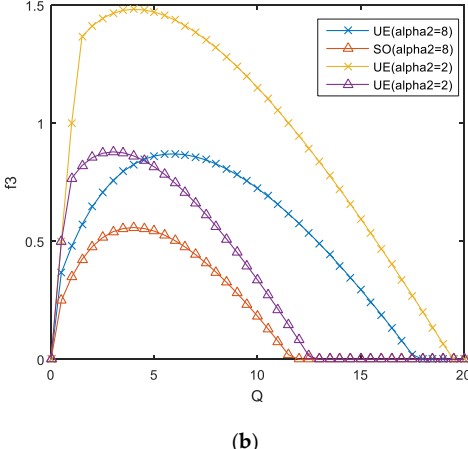

(**b**)

**Figure 11.** Sensitivity of parameter $\alpha_2$: (**a**) The effects of $\alpha_2$ on $C_r$-$C_0$; (**b**) The effects of $\alpha_2$ on $f_3$.

It can be seen from Figure 11b that when $\alpha_2$ decreases, f3 increases, which means that the traffic flows into the new roads, and the traffic growth is more obvious under the UE conditions. The irrational choice of new roads leads to congestion and the increase of the total cost in some intervals, and the possibility of the paradox happening is increased.

### 5.2.4. Sensitivity of Parameter $\beta_2$

$\beta_1$ and $\beta_2$ are the influence parameters of traffic flow on time, reflecting the traffic commuting efficiency. Figure 12a shows that when the commuter efficiency of the new road increases ($\beta_2$ decreases), the non-paradox interval decreases and even disappears under the UE condition. There are two main reasons for this phenomenon: (1) A lot of traffic flows irrationally pour into new expressways, especially under the condition of UE; (2) in the assumptions of this paper, the commuting capacity of the new road is consistent with the efficiency of the non-bottleneck section in the original road. When $\beta_2$ decreases, the commuting efficiency of the original road will also increase, so the possibility of the paradox appearing will increase. As $\beta_2$ is very small, the paradox will always appear.

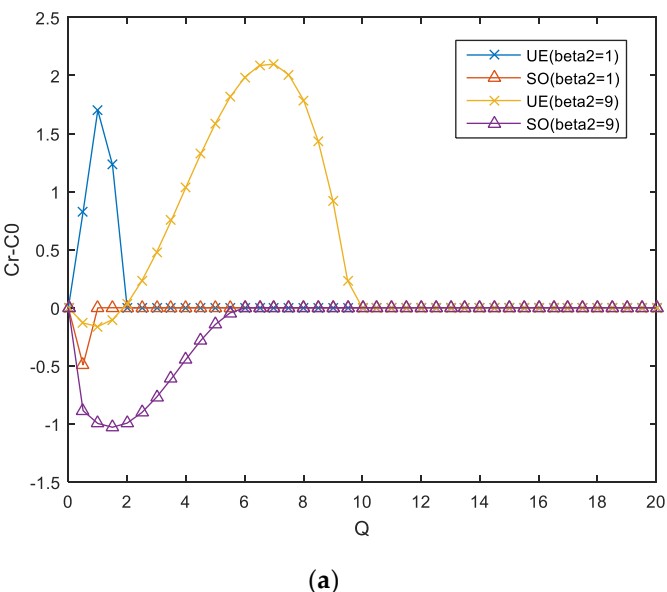

(**a**)

**Figure 12.** *Cont.*

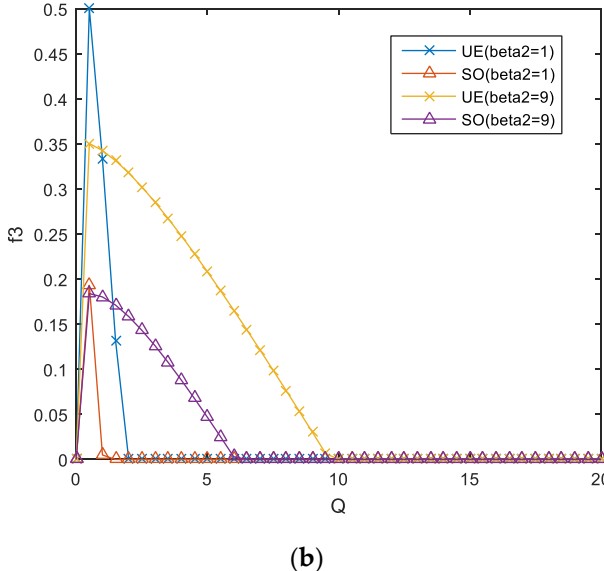

(b)

**Figure 12.** Sensitivity of parameter $\beta_2$: (**a**) The effects of $\beta_2$ on $C_r$-$C_0$; (**b**) The effects of $\beta_2$ on $f_3$.

## 6. Discussion

From the results of the mathematical derivation and the sensitivity analysis, the following observations can be made.

Firstly, the effects of $h_2$ and $\theta$ reflect the direct impact of health damage on the traffic paradox, while the sensitivity of $\alpha$ and $\beta$ reflects the indirect impact of health damage on the traffic paradox. Whether directly or indirectly, the emissions generated in the process of transportation and the impact on human health will have positive effects on the comprehensive cost of the entire transportation network and the formation of the traffic paradox. First of all, during the optimization of the whole traffic network, to improve the operation efficiency of new roads, we should fully consider the impacts of different traffic modes and road characteristics on human health and choose more reasonable methods of transportation. Some transport modes with low emissions, low damage to human health, and higher commuting efficiency should be adopted, such as urban rail transit, new energy vehicles, high-speed roads, etc. The use of more environmentally friendly and clean energy should be encouraged, such as by replacing fuel vehicles with electric vehicles and encouraging the use of more environmentally friendly emission standards for fuel vehicles. Although the emissions of motorcycles are not significant, human exposure is very severe and the health damage is relatively high. So, motorcycles should be limited in urban traffic. Secondly, it is necessary to carry out environmental governance on roads to reduce the health damage caused by particles and pollutants in traffic. For example, using asphalt roads instead of cement pavement can effectively reduce the amounts of PM2.5 and PM10 and increase road greening. Third, the publicity and popularization of the impact of traffic on human health need to be increased. Traffic participants can more clearly understand the impact of the traffic process on human health damage, which could have a positive role in optimizing the traffic network and reducing the occurrence of the Braess paradox. People who live in different urban areas may have different perceptions of the health risk caused by pollution due to differences between their surroundings and the environmental background. In addition, people from different social classes may differ in their perceptions, even if they live in the same region, due to differences in education and awareness. Therefore, it may be necessary to develop different transportation plans for different areas of the same city, since the conditions that determine whether the Braess paradox will occur and the conditions under which the system's performance can be improved differ greatly between levels of health damage.

Second, the SO condition is an obviously better traffic network equilibrium condition than the UE condition, but the SO condition puts forward higher requirements for practice. First of all, SO conditions

need to comprehensively grasp the information of the whole traffic network and reasonably guide the traffic flow, which requires management departments to make more efforts in the informatization of the traffic network. With the popularization of the Internet, GPS, and other information technologies, the traffic information in the traffic network can be better understood by the traffic participants, which also provides great help for overcoming the Braess paradox. Secondly, the irrationality of the UE conditions lies in a large number of irrational traffic influxes into new roads, which greatly increases the comprehensive traffic cost and the possibility of the traffic paradox. Therefore, in terms of traffic flow guidance, the management department can use flow restriction, diversion, and other methods to guide traffic participants to choose roads reasonably.

## 7. Conclusions

This study is the first attempt to expand analyses of the Braess paradox to account for more than just the value of time. Specifically, we extended the traditional analysis, which only accounts for the VOT, to account for the effects of health damage on route choice. We computed a total comprehensive cost of travel that combines the effects of both factors. We then identified the range of travel demands under which the new paradox occurred under UE conditions and determined whether the system's performance could be improved under SO conditions. Numerical analysis showed that whether the new paradox occurred depended on the parameter values for the functions that define the health damage and the link travel time, which is consistent with previous research.

However, in contrast to previous research, which only addressed the VOT, the new paradox is more likely to occur because the range of travel demand levels for which it occurs increases due to changes in the lower demand value at which the paradox will begin to occur. However, the range of travel demands for which the system's performance can be improved also increases. Moreover, high health damage significantly affects route choice and the distribution of traffic flows. As the awareness of health risk grows, travelers will increasingly emphasize their health over travel time. Therefore, policymakers will need to optimize the network's operation under SO conditions if the total travel demand remains low or moderate.

This paper has several limitations. First, we made many simplifying assumptions in developing our models; for example, we assume that there are only two paths in the model. These simplifications should be eliminated in future research to account for a wider and more realistic range of conditions. Second, we mainly used mathematic models to analyze the impact of air pollution on the Braess paradox; some empirical studies should be done by using data within an authentic context in future. Third, high health damage significantly affects route choices and traffic flow distribution, but the health damage we define in the paper depends on the traffic flows along links, the travel time, and the perception factor. It is somewhat simple. The health damage should be more accurately defined in the future.

**Author Contributions:** Conceptualization, Z.C. and L.Z.; methodology, Z.C.; software, Z.C.; validation, Z.C. and L.Z.; formal analysis, H.L.; writing—original draft preparation, Z.C.; writing—review and editing, L.Z.; project administration, Z.C.; funding acquisition, Z.C. All authors have read and agreed to the published version of the manuscript.

**Funding:** This research was funded by the Zhejiang Natural Science Foundation, grant number LY18G020008, the Major Project of Center for Research in Regional Economic Opening and Development of Zhejiang, grant number 16JDGH011, and the Postdoctoral fund in China, grant number 2017M621489. Zhejiang Province Soft Science Foundation of China grant number 2019C35006, General projects of Humanities and social sciences of the Ministry of Education grant number 20YJCZH005.

**Conflicts of Interest:** The authors declare no conflict of interest.

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
