# Peer review of "A Transportation Network Paradox: Consideration of Travel Time and Health Damage due to Pollution"

_sustainability, doi:10.3390/su12198107_

Round 1

Reviewer 1 Report

The work is interesting, to highlight different aspects:

The authors apply an appropriate methodology at the level of the journal.

The authors present an interesting conceptual model.

The study region is suitable for research.

The article is interesting.

The authors present an interesting study at the statistical level.

I present below the recommendations for the authors:

The article has an outdated bibliography. The literature review is not adequate. I recommend that the authors update the bibliography. This bibliography can be added in a section called “literature review”.

I recommend restructuring the article. The introduction section should be shortened. Part of the introduction of the addition to a section called “literature review”. This literature review section should be expanded and updated. In the development of the work it is quite broad. The next section could include an introduction in a simple way, explaining the methodology, how this complex section is structured and then adding the other information. Before the conclusions section add a section called discussion.

In the conclusions section, the authors should include future lines of research. And also the authors should include in conclusions the limitations of the study.

Reviewer 2 Report

The claim that there is no study on route choice and pollution effects is not true. There are studies of route choice that consider not only time and pollution costs but also slopes and access to facilities along the way.

In any case vehicle pollution is highly correlated with congestion and hence time value of travel. Hence, there is little need to consider pollution separately from congestion.

Reviewer 3 Report

This study focuses on a transportation network paradox that considers both travel time and health damage caused by pollution. I think the paper fits well the scope of the journal and addresses an important subject. However, a number of revisions are required before the paper can be considered for publication. There are quite a few weak segments in the paper. These weak segments of the paper must be strengthened. Below please find more specific comments:

*Page 1: The title seems to be pretty long. I suggest using “A Transportation Network Paradox: Consideration of Travel Time and Health Damage due to Pollution” or something in that fashion.

*Page 1: The authors start the introduction section with a discussion regarding urban cities and associated environmental issues. I recommend for the authors to strengthen the discussion regarding the global environmental problems first before discussing the issues that are associated with urban cities specifically. In particular, I recommend adding several sentences that discuss different sectors (e.g., construction, transportation, shipping, manufacturing, etc.) that are responsible for emissions of CO2 and other harmful gases across the world as well as the associated negative consequences. Such discussion will strengthen the introduction section. This discussion should be supported by the relevant references, including the following:

Lin, B. and Xu, M., 2018. Regional differences on CO2 emission efficiency in metallurgical industry of China. Energy policy, 120, pp.302-311.

Dulebenets, M.A., Moses, R., Ozguven, E.E. and Vanli, A., 2017. Minimizing carbon dioxide emissions due to container handling at marine container terminals via hybrid evolutionary algorithms. IEEE Access, 5, pp.8131-8147.

Andersson, F.N., Opper, S. and Khalid, U., 2018. Are capitalists green? Firm ownership and provincial CO2 emissions in China. Energy policy, 123, pp.349-359.

Abioye, O.F., Dulebenets, M.A., Pasha, J. and Kavoosi, M., 2019. A vessel schedule recovery problem at the liner shipping route with Emission Control Areas. Energies, 12(12), p.2380.

Lo, P.L., Martini, G., Porta, F. and Scotti, D., 2018. The determinants of CO2 emissions of air transport passenger traffic: An analysis of Lombardy (Italy). Transport Policy.

After this discussion it would be logical to start the discussion regarding urban cities and associated environmental issues specifically.

*Page 2 lines 77-80: Please pay attention to presentation of references. In particular, in many cases the square brackets from references are merged with the words (e.g., “demand[10,11]” should be replaced with “demand [10,11]”). Please address this issue throughout the entire manuscript.

*Page 2 line 81: “Literatures have discovered” does not sound well to me. I suggest using “Previous studies have discovered”.

*Page 5-6: The authors present different mathematical relationships. Please provide supporting references where applicable, so the interested readers will be able to find more details in the alternative sources.

*Page 7 line 258: “system optimization conditions” should be replaced with “System optimization conditions”, since it is a name of section 2.3.

*Page 11: Please add a few sentences to justify selection of the study area (i.e., why this particular city and not the others?).

*Page 18 line 533: “From the results of the sensitive analysis, we can see that” can be replaced with “From the results of the sensitive analysis, the following observations can be made:”.

*The conclusions section should be expanded. More specifically, please expand on limitations of this study and how they will be addressed as a part of future research.

Round 2

Reviewer 1 Report

Congratulations to the authors.
The authors have added the requested changes.
Best regards,

Author Response

Thank you for your hard work.

Reviewer 2 Report

Paper requires extensive editing as there are numerous grammatical errors. It can also be shortened considerably by focusing on the key issues instead of trying to do too many things. Finally, the math is cumbersome and can be substantially improved by removing unnecessary subscripts, simplifying expressions, etc. For example, many pages can be reduced by writing the equations as, along a route,

Time of travel: t = a + bV, where a = constant, V = traffic volume;

Damage to health: e = hVt, where h is a constant;

Cost of travel C = ct + de, where c and d are constants.

The equilibrium condition for the network is C is the same for all routes. An even simpler expression is C = f(V, t) = AVrts where A is a constant, r and s are parameters. This specification is superior tot he linear specification in the paper, and does away with many unnecessary pages. It is also superior from an economic point of view.

Reviewer 3 Report

The authors took seriously my previous comments and made the required revisions in the manuscript. The quality and presentation of the manuscript have been improved. Therefore, I recommend acceptance.

Author Response

Thank you for your hard work.